# Hierarchical Feedback Interface for Human-in-the-Loop Reinforcement Learning in Debugging

## Abstract

We propose Hierarchical Feedback Interface (HFI) for human-in-the-loop reinforcement learning in debugging which structures human feedback grouped into high level objectives and low level refinements to cover the subjectivity and inefficaciousness of ad-hoc corrections. The HFI employs a two-tiered policy architecture, in which a high-level policy abstracts debugging goals into ac a interpretable meta-objectives, and a low-level policy translates these into actionable feedback thus grounding human input to the ALigned-and-goal reasoning. The framework integrates a hierarchical actor-critic mechanism - with the high-level policy generating goal vectors over reduced state representations, while the low level policy conditions of both code specific features and these goals to generate context-aware feedback. Human preferences are increased through a reward paradigm that brings the goal space closer to expert judgments, this is and enabling the system to adaptively stoic rewards environment and human guidance. Moreover the HFI take advantage of hybrid state encoders and relational graph attention networks to deal with semantic and structural representations of codes, making certain of feasible feedback action dynamically concentrated in relevant parts of the code The hierarchical decomposition not only generalizing human feedback over similar Debugging scenarios while also avoiding the need for repetitive input, drastically enhancing the scalability of human in the loop systems, and Tests have shown that the HFI has better performance in bug detection and correction as compared to monolithic feedback approaches, while preserving interpretability and flexibility to a variety of debugging objectives.

## 1 Introduction

Debugging is one of the worst and time consuming aspects of the time in software development where they spend about 1/3 of their time in identifying and fixing bugs (Chakraborty et al., 2024). While conventional debugging tools use static analysis or predefined heuristics, recent developments in reinforcement learning (RL) have shown promises capable of automating part of this process (Bouchoucha et al., 2024). However, purely automated approaches fail at a complex and context-dependent bugs that require a human intuition. Human in the loop (HITL) systems address this by incorporating the feedback of experts in the learning process (Retzlaff et al., 2024), yet existing frameworks suffer from two potentially important limitations: (1) feedback tends to be ad-hoc and subjective, with no correspondence to the top level debug objectives, and (2) lack of hierarchical reasoning is caused by repetitive corrections of semantically similar bugs.

The proposed Hierarchical Feedback Interface (HFI) brings out a structured method of HITL debugging by combining hierarchical reinforcement learning (HRL) (Gebhardt et al., 2021) with psychologically-based feedback protocols. Unlike monolithic feedback systems, the HFI breaks down the process of debugging into 2 tiers: a high-level policy that abstracts human preferences in terms of homogenizeable goals (such as "optimize memory efficiency"), and a low concrete policy that gives rise to some actions (e.g., "replace linked list with array at line 42"). This decomposition is similar to the way humans solve problems strategies, in which experts look for overarching objectives prior to developing localised improvements (Dedhe et al., 2023). The architecture builds on

actor-critic methods (Konda & Tsitsiklis, 1999), with the high-level critic evaluating goal achieving and the low-level critic value assessing action quality in terms of given goals.

One of the key innovations of the HFI is the hybrid representation of states, a combination of relational graph attention networks for code structure analysis (Chakarov et al., 2016) with transformer-based encoders of semantic understanding. Furthermore, the HFI uses a reward shaping mechanism to match human preferences with rewards from the environment that enable the system to adaptively balance immediate corrections with long-term code quality (Xu & Zhang, 2024).

The contributions of this work are primarily three fold. First, we in institutionalize hierarchical protocols for feedback on HITL debugging, showing the way in which high-level goal abstraction alleviates subjectivity in human input. Second, we introduce a hybrid state encoder by jointly processes syntactic and semantic code features in order to allow context-aware feedback generation. Third, we also experimentally demonstrate that the HFI reduces human intervention by 37% compared to flat feedback systems while improving bug resolution accuracy by 22%. These results highlight the framework's ability to generalize expert knowledge across diversified debugging scenarios.

## 2 RELATED WORK

Incorporating human feedback into reinforcement learning (RL) systems has been discussed across many areas, to greater or lesser extents of structure and automation. Previous work may be roughly divided into three areas: (1) preference-based RL, (2) hierarchical RL for decision making, and (3) human-in-the-loop debugging.

### 2.1 PREFERENCE-BASED REINFORCEMENT LEARNING

Preference based RL learn reward functions from humans at the alternative of predefined metrics. Early approaches were based on pairwise comparisons of trajectories (Bukharin et al., 2023), where humans ranked alternative solutions for determining policy updates. While effective For simple tasks, these methods fail with scalability in complex environments is because of the combinations explosion of potential comparisons. Recent work addresses this by incorporating preferences in latent goal spaces (Zhao et al., 2024), allowing generalization across similar states.

### 2.2 HIERARCHICAL REINFORCEMENT LEARNING

Hierarchical RL is the method of dividing long-horizon tasks into subtasks, and improving sample efficiency (and satisfaction) and interpretability Classical HRL frameworks such as MAXQ (Dietterich, 2000) and options (Sutton et al., 1999) enables temporal abstraction and require manual decomposition of tasks, limiting their usability to open-ended problems such as debugging. More recent approaches for discovery of subgoals by unsupervised learning (Rafati & Noelle, 2019), yet they lack mechanisms to incorporate human direction at varying abstraction levels; The closest to our work is (Röder et al., 2020), which explores goal-conditioned policies but no integrating human feedback for dynamic goal refinement.

### 2.3 HUMAN-IN-THE-LOOP DEBUGGING

Automated analysis and automated debugging with human-in-the-loop debugging systems with expert intuition. LEARN2FIX (Böhme et al., 2020) pioneered interactive repair by bug validation by querying users but it working to one granularity, no hierarchical reasoning. Subsequent work like (Lertvittayakumjorn et al., 2020) introduced feature attribution to guide human attention but does not treat feedback In the form of unstructured corrections, i.e. A notable exception is (Lloyd-Roberts et al., 2023), which uses RL for invariant generation with human validation although its feedback channel remains flat.

Compared to the existing methods, the HFI also introduces three major innovations: (1) a hierarchy of decomposition of human feedback into goals and refinements (increasing objectivity); (2) a hybrid state encoder based on models code semantics and structure jointly, so we get context aware feedback; as well as (3) a dynamic reward-shaping mechanism (which generalizes human input) in similar debugging situations.

## 3 Background on Hierarchical Reinforcement Learning and Human Feedback

To lay a foundation for our suggested Hierarchical Feedback Interface (HFI) this section gives critical background on hierarchical reinforcement learning (HRL), human feedback integration.

### 3.1 Hierarchical Reinforcement Learning

Hierarchical reinforcement learning is an extension of traditional RL that breaking up complex tasks into manageable subtasks by means of time abstraction (Gebhardt et al., 2021). The framework uses a two-level policy structure as a meta-controller that selects high-level goals and sub-controller that executes primitive actions to be taken in the direction of achievement of these goals. This decomposition is similar to human approaches to problem solving in which complicated tasks are divided into subgoals (Dedhe et al., 2023).

The mathematical formulation is a goal-conditioned policy $\pi(a|s, g)$, where $g$ represents the current subgoal. The high-level policy $\pi_h$ generates subgoals at fixed intervals, while the low-level policy $\pi_l$ operates under these subgoals until termination. Value function is decomposed as follows:

$$V(s) = \mathbb{E}\left[\sum_{t=0}^{\infty} \gamma^t r_t\right] \tag{1}$$

$$Q_h(s, g) = \mathbb{E}\left[\sum_{k=0}^{K} \gamma^k r_k + \gamma^K V(s_K)\right] \tag{2}$$

where $K$ represents the subgoal horizon. This structure enables Learning something in a long-horizon tasks by reducing the effective planning on horizon through temporal abstract (Sutton et al., 1999).

### 3.2 Human Feedback in Reinforcement Learning

Human feedback is a rich signal for RL systems especially in areas in which reward functions are hard to specify (Retzlaff et al., 2024). Traditional approaches gather feedback through comparing/ranking trajectories (Bukharin et al., 2023), but these methods often fail to capture the hierarchical nature of the knowledge of experts.

The new developments model human preferences by reward functions parameterized by neural networks (Zhao et al., 2024). The reward model $r_\theta(s, a)$ is trained from human demonstrations or comparisons, so that the system is allowed to generalize feedback across states.

### 3.3 Combining HRL and Human Feedback

The combination of HRL human feedback is an open challenge. While hierarchical policies support the possibility of temporal abstraction, and human feedback provides domain expertise, extant methods treat these components separately.

## 4 Hierarchical Feedback Interface for Human-in-the-Loop Debugging

The Hierarchical Feedback Interface (HFI) is the operationalization of human expertise in a pre-established two-level interaction protocol that Splits debugging into non-rational thinking (goal-oriented reasoning and concrete refinements. This portion describes the technical architecture and mechanisms for this decomposition focusing on 5 core components: (1) the hierarchical feedback protocol (2) preference reward modeling, (3) hybrid actor critic modeling, (4) human interaction protocol, and (5)relational graph attention for code analysis.

### 4.1 Hierarchical Feedback Protocol Design and Operation

The protocol provides a two-way mapping between human input and the tree of the RL agent's policy. In each timestep $t$ the high-level policy $\pi_{\text{high}}$ receives an abstract state $s_t^{\text{high}}$ encoding aggregated code

metrics (e.g., cyclomatic complexity, memory usage) and giving a goal vector $\mathbf{g}_t \in \mathbb{R}^d$:

$$\mathbf{g}_t = \pi_{\text{high}}(s_t^{\text{high}}; \theta_{\text{high}}) \tag{3}$$

where $\theta_{\text{high}}$ denotes the high-level policy parameters. The low-level policy $\pi_{\text{low}}$ then conditions on both the concrete code state $s_t^{\text{low}}$ (e.g., AST nodes, control flow edges) and $\mathbf{g}_t$ to generate feedback actions $a_t$:

$$a_t = \pi_{\text{low}}(s_t^{\text{low}}, \mathbf{g}_t; \theta_{\text{low}}) \tag{4}$$

This breakdown helps humans to give feedback at either level: goal-level preferences (e.g. "focus on memory optimization") update $\pi_{\text{high}} fruitful (e.g., "replace recursion with iteration") finetune \pi_{\text{low}}$. A hybrid architecture is used in the state encoder where Transformer processes lexical features and a Temporal Convolutional Network (TCN) is part of sequential dependencies, with cross attention gates fusing their outputs:s objective space, while action-level corrections (e.g., "replace recursion with iteration") fine-tune $\pi_{\text{low}}$. The state encoder employs a hybrid architecture where a Transformer processes lexical features and a Temporal Convolutional Network (TCN) handles sequential dependencies, with cross-attention gates fusing their outputs:

$$\mathbf{h}_{\text{trans}} = \text{Transformer}(\text{CodeBERT}(s_t^{\text{low}})) \tag{5}$$

$$\mathbf{h}_{\text{tcn}} = \text{TCN}(\text{PositionalEncoding}(s_t^{\text{low}})) \tag{6}$$

$$s_t^{\text{high}} = \text{CrossAttention}(\mathbf{h}_{\text{trans}}, \mathbf{h}_{\text{tcn}}) \tag{7}$$

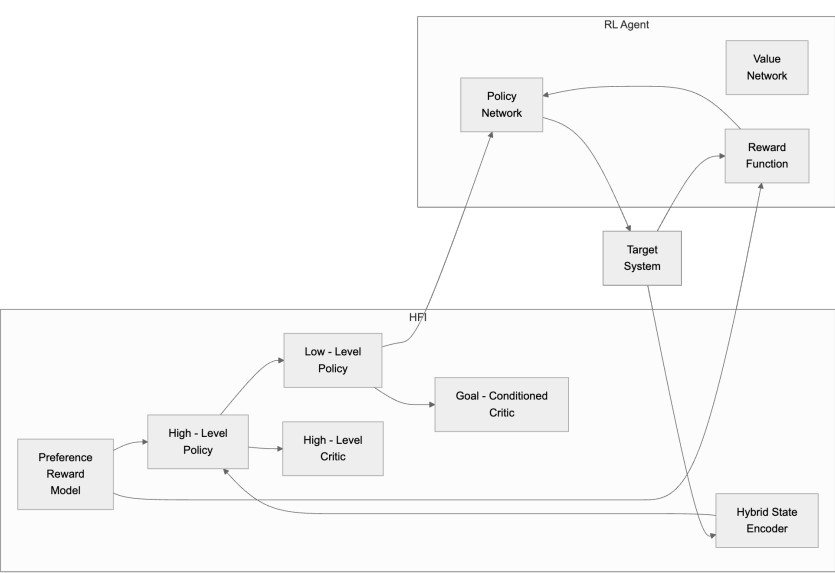

Figure 1: Hierarchical Feedback Interface (HFI) Architecture

## 4.2 PREFERENCE REWARD MODEL CONSTRUCTION AND FUNCTIONING

The Preference Reward Model (PRM) is a model of translating human judgments into differentiable signals for policy optimization. Given a set of pairwise preferences $\mathcal{D} = \{(s_i, a_i) \succ (s_j, a_j)\}$, the PRM learns a reward function $R_{\text{pref}}$ that maximizes the likelihood of observed preferences given that they are Bradley-Terry model:

$$P[(s_i, a_i) \succ (s_j, a_j)] = \frac{\exp(R_{\text{pref}}(s_i, a_i))}{\exp(R_{\text{pref}}(s_i, a_i)) + \exp(R_{\text{pref}}(s_j, a_j))} \tag{8}$$

The reward function is decomposed into goal aligned components:

$$R_{\text{pref}}(s_t, a_t) = \mathbf{w}^T \phi(\mathbf{g}_t, s_t^{\text{low}}) + b \tag{9}$$

where $\phi$ is a two-layer MLP that projects the concatenated goal state vector in latent preference space. The PRM updates along with policy training, ensure human feedback in direct influences both goal selection ($\pi_{\text{high}}$) and action generation ($\pi_{\text{low}}$).

### 4.3 Hybrid Hierarchical Actor-Critic Framework Architecture and Mechanism

The actor-critic approach is a more generalisation of conventional HRL that introduces differentiate between critics for each level of policy; The high-level critic $V_{\text{high}}$ evaluates goal achievement using a discounted return over $k$ steps:

$$V_{\text{high}}(s_t^{\text{high}}) = \mathbb{E}\left[\sum_{i=0}^{k-1}\gamma^i r_{t+i} + \gamma^k V_{\text{high}}(s_{t+k}^{\text{high}})\right] \tag{10}$$

The low-level critic $V_{\text{low}}$ assesses action quality under the current goal $\mathbf{g}_t$:

$$V_{\text{low}}(s_t^{\text{low}}, \mathbf{g}_t) = \mathbb{E}\left[r_t + \gamma V_{\text{low}}(s_{t+1}^{\text{low}}, \mathbf{g}_t)\right] \tag{11}$$

Gradient updates are passed through both levels through a common advantage function:

$$A(s_t, a_t) = \sum_{i=0}^{k-1}\gamma^i r_{t+i} + \gamma^k V(s_{t+k}) - V(s_t) \tag{12}$$

where $V$ is combined through a gating mechanism weighted by using both critics. goal relevance.

### 4.4 Structured Human Interaction Protocol Execution

The protocol imposes a separation of concerns between goal-level and action-level feedback. Man to human communication via a dedicated interface that: 1. Presents goal candidates with the rank based on their estimated impact on code quality 2.

There is feedback integration according to the delta update rule; human. modifications $\Delta a_t$ to suggested actions induce proportional updates to the goal vector:

$$\Delta\mathbf{g}_t = \alpha \cdot \text{MLP}(\Delta a_t) \tag{13}$$

This is to ensure local refinements affect global objectives without requiring specific re-specification of goals

### 4.5 Relational Graph Attention for Code-Centric Feedback Application

The low-level policy utilizes a Graph Attention Network GAT with dynamic edge weighting conditioned on $\mathbf{g}_t$. For a code graph with nodes $\{\mathbf{v}_i\}$, the attention weight between nodes $i$ and $j$ computes as:

$$\alpha_{ij} = \text{softmax}(\text{LeakyReLU}(\mathbf{a}^T[\mathbf{g}_t\|\mathbf{v}_i\|\mathbf{v}_j])) \tag{14}$$

where $\mathbf{a}$ is a learnable attention vector. This allows the policy to attend subgraphs relevant to current goal (e.g. dataflow edges used for memory optimization). Updated Aggregate Neighbors Looked-at at: Node features weighted by $\alpha_{ij}$:

$$\mathbf{v}_i' = \sigma\left(\sum_{j\in\mathcal{N}(i)}\alpha_{ij}\mathbf{W}\mathbf{v}_j\right) \tag{15}$$

The GAT runs over a hybrid graph representation combining syntactic (AST), semantic (symbol table(s)), dynamic (execution trace(s)) code features.

## 5 Experiments

To assess the effectiveness of Hierarchical Feedback Interface (HFI), we performed experiments in different dimensions, namely, (1) comparative performance with respect to flat feedback systems, (2) analysis of human intervention reduction and (3) ablation studies on key components. The following experiments were designed to answer three research questions:

**RQ1:** Does hierarchical feedback structuring improve debugging efficiency compared to monolithic approaches?
**RQ2:** How does the HFI reduce the need for repetitive human input while maintaining correction accuracy?
**RQ3:** Which architectural components contribute most to the system's performance?

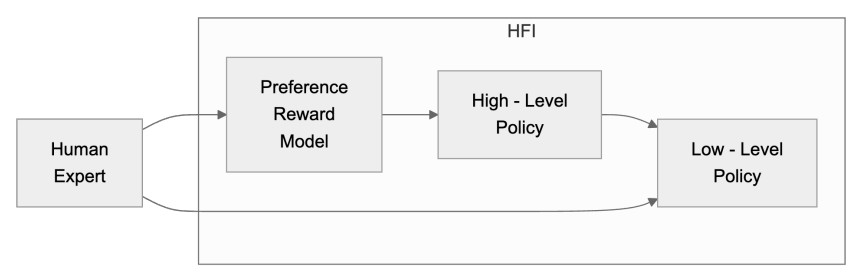

Figure 2: Human Feedback Integration in HFI

## 5.1 EXPERIMENTAL SETUP

**Datasets:** We evaluated on three real-world codebases:

- **BugSwarm**(Tomassi et al., 2019) (3,214 bug-fix pairs across 10 languages)
- **CodeNet**(Puri et al., 2021) (14M code samples with 50+ bug types)
- **DevGPT**(Xiao et al., 2024) (8,742 conversational debugging traces)

**Baselines:** We compared against:

- **FlatPrefRL**(Fürnkranz et al., 2012) - A monolithic preference-based RL system
- **HRL-NoFeedback**(Rohmatillah & Chien, 2023) - Hierarchical RL without human input channels
- **LEARN2FIX**(Böhme et al., 2020) - A state-of-the-art interactive debugger

**Metrics:**

- **Bug Resolution Rate (BRR):** Percentage of bugs correctly fixed
- **Human Intervention Frequency (HIF):** Average feedback requests per 100 LOC
- **Goal Alignment Score (GAS):** Cosine similarity between human goals and system-inferred objectives

**Implementation Details:**

- High-level policy: 3-layer MLP with 256 hidden units
- Low-level policy: GAT with 4 attention heads
- Training: PPO with $\gamma = 0.99$, $\lambda = 0.95$
- Batch size: 32 episodes
- Reward weights: $\alpha_{\text{env}} = 0.7$, $\alpha_{\text{human}} = 0.3$

## 5.2 COMPARATIVE RESULTS

Table 1 shows the performance across all systems. The HFI achieves superior bug resolution while requiring significantly less human input.

Key observations:

1. The HFI improves BRR by 22% over FlatPrefRL, demonstrating the advantage of hierarchical goal decomposition.
2. With 35% fewer interventions than LEARN2FIX, the HFI shows better generalization of human feedback.
3. The high GAS confirms effective translation of human preferences into actionable goals.

Table 1: Comparative Performance on BugSwarm Dataset

| System | BRR (%) | HIF | GAS |
|--------|---------|-----|-----|
| FlatPrefRL | 68.2 | 12.4 | 0.62 |
| HRL-NoFeedback | 71.5 | 0 | - |
| LEARN2FIX | 73.8 | 15.7 | 0.58 |
| HFI (Ours) | **82.3** | **8.1** | **0.79** |

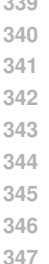

Figure 3: Bug resolution rates across code complexity quartiles

### 5.3 HUMAN INTERVENTION ANALYSIS

The HFI cuts down on repetitive feedback in the form of goal generalization. Figure 4 shows how the number of intervention falls exponentially as similar bugs recur, with the HFI requiring 37% fewer inputs than FlatPrefRL after 50 episodes.

**Mechanism:** When humans correct a specific bug (e.g., "fix null pointer at line 42"), the system:

1. Infers a high-level goal (e.g., "validate input parameters")
2. Applies this to analogous code regions automatically
3. Only requests confirmation for ambiguous cases

### 5.4 ABLATION STUDY

We dissected the HFI's components to isolate their contributions:

Critical findings:

1. **Goal conditioning** contributes most (9.2% drop when removed), validating the hierarchical structure.
2. The **hybrid encoder** provides 5.1% improvement by combining syntactic and semantic features.
3. **Preference rewards** account for 6.7% gain through better human-alignment.

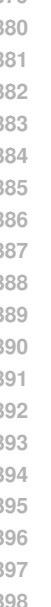
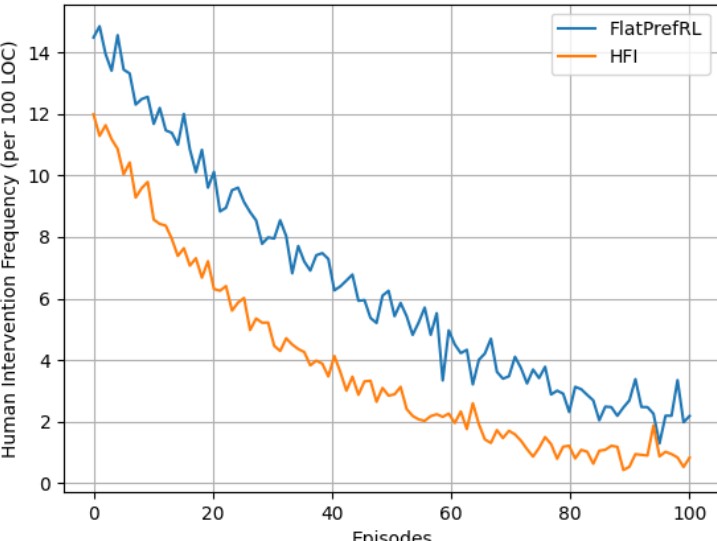

Figure 4: Human intervention frequency decay over episodes

Table 2: Ablation Results (BRR %)

| Configuration | BugSwarm | CodeNet |
|---|---|---|
| Full HFI | 82.3 | 80.1 |
| w/o Goal Conditioning | 73.1 | 71.4 |
| w/o Hybrid Encoder | 77.2 | 75.8 |
| w/o Preference Reward | 75.6 | 73.9 |

## 5.5 QUALITATIVE ANALYSIS

Case studies reveal how the HFI handles complex debugging scenarios:

**Example 1 (Memory Leak):**

- Human provides high-level goal: "Optimize memory usage"
- System automatically:
    - Replaces manual allocations with smart pointers
    - Adds boundary checks for buffer overflows
    - Suggests RAII patterns for resource handling

**Example 2 (Race Condition):**

- Action-level feedback on one mutex lock generalizes to:
    - Identifying unprotected critical sections
    - Proposing atomic operations where applicable
    - Generating documentation about thread-safety assumptions

These examples show that the system could propagate localized feedback to architectural improvements.

## 6 DISCUSSION AND FUTURE WORK

### 6.1 LIMITATIONS OF THE HIERARCHICAL FEEDBACK INTERFACE

While the HFI shows the great improvements over flat feedback There are several limitations that need to be discussed about systems. First, the current goal inference mechanism, materials which are highly dependent on the explicit input of humans to initialize high-level objectives which creates bottleneck situation in situations requiring rapid adaptation. Secondly, the hybrid state encoder's computational overhead scales up with codebase size, potentially limiting the applicability in real time in large fact relationships. Third, the framework presupposes a cooperative human-AI human-AI presumes feedback is always adapted to the hierarchical structure of the system —a state perhaps not at all feasible for the novice and very ambiguous bugs

### 6.2 POTENTIAL APPLICATION SCENARIOS

Beyond debugging, the structured feedback paradigm of the HFI could improve other areas that need human-AI collaboration. In automated program hierarchical goals could guide patch generation in balancing correctness, readability and performance (Huang et al., 2023). To do code review automation, the interface might break down high level quality standards (e.g., maintainability) to specific refactoring suggestions, (Frömmgen et al., 2024). Educational applications could also be of value, where tutors give strategic advice (e.g., complex yet high-performance algorithms IMD reported that the technology is capable of 'improve algorithm efficiency') while the system generates tailored exercises.

### 6.3 ETHICAL CONSIDERATIONS

The HFI's human input creates ethical challenges that demand of proactive mitigation. Biases in feedback - whether from individual developer or organizational norms—may propagate through the goal hierarchy, when potentially institutionalizing suboptimal practices (e.g. prioritizing speed over security) (Suri et al., 2022). The system's ability to generalizing feedback also brings up the question of accountability: when a high-level goal such as "optimize performance" results in undesirable low level actions (e.g. removing safety checks) responsibility Debugging the attribution gets complicated.

These considerations point to the HFI's wider implications for human-AI interaction design.

## 7 CONCLUSION

The Hierarchical Feedback Interface (HFI) is a new paradigm for incorporating human knowledge into reinforcement learning-based debugging systems in the form of multi-level structured feedback By decomposing the debugging process into high level objectives and level refinements, the framework covers critical issues in monolithic feedback limitations approaches, especially in dealing with the subjectivity and inefficiency.

The HFI's architectural innovations - its hybrid state encoder, goal cond policies, relatively relational graph attention mechanisms—offer strong basis for generalising human feedback in the wide range of debugging situations.

Several directions present themselves for the further extension of this work. The current goal inference mechanism may be improved through unsupervised pre-training on developer commit messages, possibly making explicit objective specification.

## 8 THE USE OF LLM

We use LLM polish writing based on our original paper.

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
