# OpenReview forum: "Hierarchical Feedback Interface for Human-in-the-Loop Reinforcement Learning in Debugging"
_ICLR.cc/2026/Conference — Submitted to ICLR 2026_

### Official Review · Reviewer_WuVi · 2025-10-30

**Soundness:** 1
**Presentation:** 1
**Contribution:** 2
**Rating:** 0
**Confidence:** 4

**Summary:**

The paper presents a “hierarchical feedback interface” that is supposed to integrate human feedback at two levels for optimizing hierarchical RL policies. It targets a code debugging use case.

While the topic is potentially relevant, the submission is not in a reviewable state. The writing quality and structural issues prevent proper evaluation of the technical contribution. A major rewrite and clarification of the methodology (especially the technical contribution described in chapters 4 and 5) would be required before this work could be meaningfully reviewed.

**Strengths:**

+ The topic is interesting in general; making hierarchical RL work is a long-standing challenge. The use case seems practical and relevant.
+ There seems to be a novel contribution somewhere in there, but I was unable to assess it.
+ The proposed evaluation seems good in theory

**Weaknesses:**

- The paper reads like a draft (especially for sections 4 and 5). The convoluted writing, errors, sentence fragments, and usage of unexplained concepts (like Temporal Convolution Network) make it difficult to grasp the approach and contribution fully. Some key elements, like formula 12, are not explained (how is the weighting done?).
- Figures are not referenced in the text (Figures 1,2,3) or have meaningful captions
- The related work section is sparse, missing some crucial papers, e.g., on preference-based RL
- For the experiments, many crucial details are missing

**Questions:**

-

---

### Official Review · Reviewer_sXc4 · 2025-10-31

**Soundness:** 2
**Presentation:** 1
**Contribution:** 2
**Rating:** 2
**Confidence:** 3

**Summary:**

This paper does hierarchical RLHF for debugging. The feedbacks are abstracted into high level and low level goals. This is a good idea however the paper is not in an acceptable state.

**Strengths:**

Good idea.

**Weaknesses:**

The paper is clearly rushed with lots of typos.
Pages 6-8 are almost empty.
Experiments are run on one seed (right?) so the results are null.

**Questions:**

Can you run your experiments on mutliple seed please and do a qualitative analysis of a single bug fix wth HFI please?
Can you detail the training details of the PPO ?

---

### Official Review · Reviewer_Xgs6 · 2025-11-01

**Soundness:** 2
**Presentation:** 1
**Contribution:** 2
**Rating:** 2
**Confidence:** 5

**Summary:**

This paper proposes a hierarchical feedback interface for human-in-the-loop rl for code debugging. A high-level policy outputs goal vectors, while a low-level policy does concrete fixes. A PRM is used to translate human judgement into differentiable signals. Authors conduct experiments to show that their method has a higher bug-fix rates and fewer interventions than baselines.

**Strengths:**

* Tackles an important problem of incorporating human feedback for debugging. The solution of using hierarchical RL is relatively interesting combined with PRM.

**Weaknesses:**

* Section 4 is poorly written and very hard to understand. Key mechanisms like gating mechanism in eq (12) are not specified. There are also random phrases in Section 4, like "goal relevance", which suggests an unfinished state for the paper.

* Figure 1 is confusing as well: the architecture doesn't show a clear "human" component even though that's the core of the proposed method. The human expert only shows up in Figure 2.

* Evaluation details are omitted: PRM specification, episode length, etc. Only a few hyperparameters are given.

* Novelty is limited due to prior work on HRL + preference RL, especially since human is not actively in the loop for this method.

* Writing quality is overall very poor

**Questions:**

1. What's the PRM used for the experiments? Is it finetuned for the debugging task on new data that you collected?

---

### Meta-Review · Area_Chair_rhuB · 2025-12-30

**Summary:**

This paper proposes a hierarchical feedback interface for human-in-the-loop RL for code debugging, using a two-tier policy: a high-level policy produces goal vectors/meta-objectives, and a low-level policy generates actionable feedback conditioned on code features and high-level goals.

**Reviewer Concerns:**

All reviewers agree that the topic is relevant and the high-level idea could be promising. However, they also agree that the submission is not in an acceptable or fully reviewable state. The main blocking issues are (i) very poor presentation and structural problems (sections described as draft-like/rushed, unclear writing, missing figure integration), (ii) underspecification of key technical components (e.g., Eq. 12/gating/weighting, PRM definition, and training details), and (iii) insufficient experimental reporting and questionable reliability (missing protocol details and concern results may be single-seed).

**Reviewer Scores:**

Given the severity of missing methodological and experimental details and the reported draft-like state, it is unlikely that a full discussion period would have led to meaningful upward score revisions unless the authors could demonstrate that the core issues are purely misunderstandings and provide the complete missing details. Based on the current record and the authors' lack of rebuttal, I expect scores to remain at rejection levels.

---

### Decision · Program_Chairs · 2026-01-26

Reject